# Joint modeling of longitudinal change in pulse rate and survival time of heart failure patients treated at Arbaminch General Hospital, Southern Ethiopia

**Belay Belete Anjullo**[1]*, **Sebisibe Kusse Kumaso**[2], **Markos Abiso Erango**[1]

**1** Department of Statistics, Arba Minch University, Arba Minch, Ethiopia, **2** Health Monitoring and Evaluation department, Alle Special Woreda, Arba Minch, Ethiopia

* belay.belete@amu.edu.et

**Data Availability Statement:** All relevant data are within the paper and its Supporting Information files.

## Abstract

### Introduction

Heart failure is a chronic progressive disease in which the heart muscle is unable to pump enough blood to meet the body's need. It is a severe health problem around the world with high re-hospitalization and death rates. The main aim of this study was to identify the factors associated with longitudinal change of pulse rate and survival time to death of congestive heart failure patients treated at Arba Minch General Hospital.

### Methods

A retrospective study design was undertaken on congestive heart failure patients admitted to the Arba Minch General Hospital from January 2017 to December 2020. Data was collected from a total of 199 patients. After evaluating the longitudinal data with a linear mixed model and the survival time to death data with cox proportional model, Bayesian joint model of both sub models was fitted in R software using JMbayes2 package.

### Results

Findings from Bayesian joint model revealed that the estimated value for the association parameter was positive and statistically significant. This indicates that there is significant evidence of an association between the mean longitudinal change of pulse rate and the risk of death. Weight of patients at baseline, gender, chronic kidney disease, left ventricular ejection fraction, New York Heart Association classification, diabetes, tuberculosis, pneumonia and family history were statistically significant factors associated with mean evolution of pulse rate of congestive heart failure patients. Left ventricular ejection fraction, etiology of congestive heart failure, type of congestive heart failure, chronic kidney disease, smoking, family history, alcohol and diabetes were found to be statistically significant factors associated with survival time to death.

**Funding:** The authors received no specific funding for this work.

**Competing interests:** The authors declare that they have no competing interests.

## Conclusion

To reduce the risk level, health professionals should give attention to congestive heart failure patients with high pulse rate, co-morbidities of chronic kidney disease, tuberculosis, diabetic, smoking status, family history, and pneumonia in the study area.

## 1. Introduction

Heart failure (HF) is a chronic progressive disease in which the heart can't keep up with its workload. It results from the failure of the heart to pump enough blood into the circulation due to ventricular a systolic dysfunction defined as left ventricular ejection fraction(LVEF)< 40% to 50% (HF with depressed ejection fraction) [1]. It is the final stage of all cardiac disorders and could be a major cause of morbidity and death [2]. With emergent urbanization, changes in lifestyle habits, and the ageing of the population, the range of causes of heart failure has also extended which results in a significant burden of both communicable and non-communicable etiologies [3].

Heart failure is a severe health problem around the world with high re-hospitalization and death rates [4]. The global re-hospitalization rate in patients with HF is over 50% [5]. As a result, heart failure affects 33 million people worldwide, or 26.4 percent of the adult population. The developed world accounts for 65.73 percent of the adult population, while developing countries account for 34.27 percent. It is estimated that by the year, there will have been a 60% increase over the year 2000 [6]. It is the world's fastest-growing cardiovascular disease, putting a significant strain on healthcare systems around the world [7]. Congestive heart failure (CHF) has become one of the top causes of death among those who have a lower quality of life and a shorter lifespan. In 2017, 960,000 new instances of congestive heart failure were identified in the United States alone, and this number is expected to rise year after year as the population ages. It is estimated that by 2030, the global prevalence will have increased by 8 million people [8].

Low and middle-income countries had more prevalent heart failure associated death than high-income countries [9]. It was reported that Sub-Saharan Africa is the one of the low-income countries in which the magnitude of risk factors associated with heart failure is increasing and heart failure has been established as a significant contributor to the burden of cardiovascular illness [10,11]. It was reported that patients with heart failure in Africa were the youngest and most likely to be in New York Health Association functional class type IV [12]. A study conducted in Sub-Saharan Africa Survey of Heart Failure identified a growing cause of heart failure from 23% to 43% in persons with heart failure [13]. It has become one of the top causes of death among those who have a lower quality of life and a shorter lifespan [14]. According to the WHO [15] report, in 2012 around 9% of all deaths in Ethiopia was due to heart failure disease.

In order to monitor the burden of congestive heart failure disease, it is required to jointly model the biomarker of the disease and survival time of patients. Hence, the main aim of this study was to identify the factors associated with longitudinal change of pulse rate and survival time to death of congestive heart failure patients treated at Arba Minch General Hospital using a Bayesian joint modeling approach.

## 2. Data and methodology

### 2.1. Cohort-based data

The data used in this study was obtained from a cohort-based retrospective study of patients diagnosed with congestive heart failure at Arba Minch General Hospital. Patients were

followed on pulse rate every month from January 2017 to December 2020 and the target population of this study includes all congestive heart failure patients under follow-up who had at least three pulse rate measurements after the first report of congestive heart failure. Patients whose medical cards were incomplete and registered during the data collection time were excluded. Therefore, a total sample of 199 patients who have full records or complete history during the study period was considered in this study. The study was carried out after getting approval on data collection from ethical committee at Arba Minch General Hospital. In addition, all study methods were performed based on relevant guidelines and regulations laid down by the Committee.

## 2.2. Variables in the study

The response variable considered in this study was the longitudinal pulse rate and survival outcome variable, where the pulse rate of the patients was measured every month for each congestive heart failure patient under follow-up, and the survival outcome variable was the time to-death of the patient under follow-up. The explanatory variables considered in this study were patients age (in year), weight(in kilogram), visiting time (monthly follow up), body temperature, blood pressure, and left ventricular ejection fraction in percent, patient gender, place of residence, smoking status of patients (smoker, nonsmoker), diabetes status of patients (present, absent), tuberculosis status of patients (positive, negative), family history of patients (present, absent), chronic kidney status of patients (present, absent), anemia status of patients (anemic, non-anemic), alcohol intake (yes, no), pneumonia status of patients (present, absent), etiology of heart failure (Valvular Heart Disease, Hypertensive heart disease, Ischemic Heart disease, Other), type of congestive heart failure patient (left ventricular, right ventricular, biventricular) and New York Heart Association classification (class I, class II, class III, class IV).

## 2.3. Methods of data analysis

Before modeling, exploratory data analysis was conducted to investigate various structures and patterns exhibited in the data set. This consists of obtaining the summary statistics such as mean and variance for pulse rate. Besides, the individual profile plots, mean structure, and variance structure plots were used to gain some insights into the data on the longitudinal outcome. While, the individual profile plots and the variance structure were used to gain insight into the variability in the data and to determine which random effects to be considered in the linear mixed model. Also, the mean structure was used to gain intuition on the time function that can be used to model the data.

**2.3.1. Joint longitudinal-survival models.** The joint model consists of two linked submodels, known as the longitudinal sub-model, and the survival sub-model was used as given subsequently.

*1. Linear Mixed Model.* Linear mixed model (LMM) is the most frequently used random effects model in the context of continuous repeated measurements from longitudinal responses when the measurements are taken on the same or related subjects at different times, in both cases, the responses are likely to be correlated [16–18]. When modeling longitudinal data, our interest is to study the association between dependent variable and a set of explanatory variables [19]. In this study, the dependent variable, pulse rate was taken on the same subject at different times with different baseline characteristic, LMM was used to model longitudinal measurement on pulse rate taken on the same subject at different time points and

the model has the form given in Eq (1):

$$\boldsymbol{Y}_i(t) = \boldsymbol{X}_i(\boldsymbol{t})\boldsymbol{\beta} + \boldsymbol{Z}_i(\boldsymbol{t})\boldsymbol{u}_i + \boldsymbol{\varepsilon}_i(t)$$
$$= m_i(t) + \boldsymbol{\varepsilon}_i(t)$$
$$m_i(t) = \boldsymbol{X}_i(\boldsymbol{t})\boldsymbol{\beta} + \boldsymbol{Z}_i(\boldsymbol{t})\boldsymbol{u}_i$$

(1)

$$\begin{cases} u_i \sim N(0, D) \\ \varepsilon_i \sim N(0, \Sigma) \\ u_1, \ldots, u_n \; and \; \varepsilon_1, \ldots, \varepsilon_n \; are \; independent \end{cases}$$

Where $\boldsymbol{Y}_i(t)$ is the $n_i \times 1$ pulse rate for the i[th] patient at time t, $\boldsymbol{X}_i$ is $n_i \times p$ dimensional design matrix of fixed predictors linking $\boldsymbol{\beta}$ to the set of longitudinal measurements of pulse rate, $\boldsymbol{Z}_i$ is $n_i \times q$ dimensional design matrix values of the random factors linking $\boldsymbol{u}_i$ to $\boldsymbol{Y}_i$ for i[th] patient, $\boldsymbol{\beta}$ is a p dimensional vector containing fixed effects, $\boldsymbol{u}_i$ is a q dimensional vector containing random effects of i[th] patient and $\boldsymbol{\varepsilon}_i$ is distributed as $N(0, \Sigma_i)$ is a vector of residual components, combining measurement error and serial correlation. Then $u_i$ is distributed as $N(0,\Omega)$, independently of each other. That is, $cov(u_i, \varepsilon_i) = 0$. Furthermore $\Sigma_i = \delta^2 I_{ni}$ is the $n_i \times n_i$. positive-definite variance-covariance matrix for the errors in subject $i$, where $I_{ni}$ denotesthe $n_i \times n_i$ identity matrix.

*2. The Cox Proportional Hazards Model.* The Cox proportional-hazards model is essentially a regression model commonly used statistical in medical research for investigating the association between the survival time of patients and one or more predictor variables [20]. A proportional hazards model proposed by [20] assumes at the hazard function $h (t, X, \gamma)$ is related to the covariates as a product of a baseline hazard $ho (t)$ and a function of covariates, has a form as shown in Eq (2):

$$h_i(t|X_i) = h_o(t)\exp(\gamma^T X_i)$$
$$= h_o(t)\exp(\gamma_1 X_1 + \gamma_2 X_2 + \cdots + \gamma_p X_p)$$

(2)

Where, $ho (t)$ is the baseline hazard function, $X = (X_1, X_2, X_3, \ldots, X_p)$ is a set of covariates from i[th] patient and $\gamma = (\gamma_1, \gamma_2, \gamma_3, \ldots, \gamma_p)$ is unknown $p$ regression parameters which measure the effect of the covariates on the risk of death. Hence, the joint model [21] that link the longitudinal response to the time-to-event process through current value parameterization has the form given in Eq (3):

$$h_i(t|m_i(t), X_i) = h_o(t)\exp(\gamma^T X_i + \alpha m_i(t)), t > 0$$

(3)

where $m_i(t) = \{m_i(s), 0 \leq s < t\}$ denotes the history of the true unobserved longitudinal process up to time point $t$ given in Eq (1), $h_o(t)$ denotes the baseline risk function, and $X_i$ is a set of baseline covariates with a corresponding vector of regression coefficients $\gamma$ and $\alpha$ *is* association parameter which quantifies the effect of the underlying longitudinal outcomes to the risk for an event. Here in this study all the parameters of joint model are estimated under a Bayesian framework using Markov chain Monte Carlo (MCMC) methods with Gibbs sampling using the JMbayes2 package in R software. The empirical results from a given MCMC analysis are not deemed reliable until the chain has reached its stationary distribution. On account of this, the term convergence of an MCMC algorithm refers to whether the algorithm has reached its target distribution. Hence, monitoring the convergence of the algorithm is essential for producing results from the posterior distribution of interest [22]. Among several ways the most popular and straight forward convergence assessment methods; time series(history) plot and

density plot [23] were used to assess whether the sample has reached its stationary distribution or not. Summary statistics (posterior mean and credible interval of posterior mean) was computed for each parameter. Finally, the importance of each of the explanatory variables is assessed by carrying out statistical tests of the significance of the regression coefficients (posterior mean) via 95% Bayesian credible interval of the posterior mean [24].

### 2.4. Ethical approval and consent

The study was carried out after getting permission from the Statistics Department, Arba Minch University. In this regard, the official letter of co-operation referenced with stat/534/2013 was written to ethical approval committee at Arba Minch General Hospital. Then, the ethical committee approved the letter and gave permission to collect data from patients' record and to use in the study. For the purpose of confidentiality, there was no links with individual patients and all data had no personal identifier. Therefore, informed consent to the patient has been waived by Arba Minch General Hospital ethical committee.

## 3. Results and discussion

### 3.1. Descriptive analysis

As indicated in Table 1, from a total of 199 congestive heart failure patients, 50.3% who received treatment were women, whereas the remaining 49.7% were men. The majority, 58.8% of the patients with congestive heart failure were from in rural areas. 127 (63.8%) of those with congestive heart failure had a family history of heart failure, whereas 72 (36.2%) had no family history of heart failure. Regarding the New York Heart Association (NYHA) class of congestive heart failure patients, among 199 patients, 5% of them were New York Heart Association class I, 11.6% were New York Heart Association class II, 29.1% were New York Heart Association class III and 54.3% were New York Heart Association class IV. Regarding history of disease status, among 199 patients, 56.3% of heart failure patients had diabetes, 35.7% had chronic kidney disease, 37.2% were smokers, 43.7% had pneumonia, 59.3% had tuberculosis, and 70.9% had anemia as comorbidities.

Similarly, out of the total 199 congestive heart failure treatment followers, about 24 (12.1%) female respondents died from treatment and the remaining were censored. On the other hand, about 19 (9.5%) of the male respondents died, and the rest of the respondents were censored. Based on the patient's residence, out of 82 urban and 117 rural patients, 22 (11.1%) and 21 (10.5%) of the respondents had an occurrence of an event, respectively (see Table 1).

Regarding continuous predictors, the average of age patients at baseline was 48.6 years with a standard deviation of 17.385 years, average weight and left ventricular ejection fraction of patients at baseline were 54.21 kilograms and 43.18% with a standard deviation of 10.93 kilograms and 13.93% respectively.

### 3.2. Exploratory data analysis

Exploratory analysis of longitudinal data attempts to find patterns of systematic variation across groups of subjects as well as characteristics of random variation that distinguish each patient.

Fig 1 depicted individual profile plot of the longitudinal pulse rate of all patients and fifteen randomly selected congestive heart failure patients by follow-up time. Hence, it can be observed that some trajectories were steeper while others were almost identical; suggesting that the slope and intercept of pulse rate readings may vary. After they begin follow-up, there seems to be a variation in pulse rate measurement over time, with the variability of the pulse

**Table 1. Summary statistics for baseline characteristics of CHF patients treated at Arba Minch General Hospital from January 2017 to December 2020.**

| Variables | Categories | Frequency (%) | Survival status | |
|---|---|---|---|---|
| | | | Event (%) | Censored (%) |
| Gender | Female | 100 (50.3) | 24 (12.1) | 76(38.2) |
| | Male | 99 (49.7) | 19 (9.5) | 80(40.2) |
| Residence | Rural | 117(58.8) | 21 (10.5) | 96(48.3) |
| | Urban | 82(41.2) | 22 (11.1) | 60(30.1) |
| Family History status | Absent | 72(36.2) | 16(8.1) | 56(28.1) |
| | Present | 127(63.8) | 27(13.6) | 100(50.2) |
| Diabetes status | Absent | 87(43.7) | 9(4.5) | 78(39.2) |
| | Present | 112(56.3) | 34(17.1) | 78(39.2) |
| Tuberculosis status | Negative | 81(40.7) | 7(3.5) | 74(37.2) |
| | Positive | 118(59.3) | 36(18.1) | 82(41.2) |
| Smoking status | Non smoker | 125(62.8) | 24(12.1) | 101(50.7) |
| | Smoker | 74(37.2) | 19(9.6) | 55(27.6) |
| Pneumonia status | Absent | 112(56.3) | 16(8.1) | 96(48.2) |
| | Present | 87(43.7) | 27(13.6) | 60(30.1) |
| Alcohol intake | No | 103(51.8) | 21(10.6) | 82(41.2) |
| | Yes | 96(48.2) | 22(11.1) | 74(37.1) |
| Anemia status | Anemic | 141(70.9) | 39(19.6) | 102(51.3) |
| | Non anemic | 58(29.1) | 4(2) | 54(27.1) |
| Chronic Kidney disease | Absent | 128(64.3) | 21(10.6) | 107(53.8) |
| | Present | 71(35.7) | 22(11.1) | 49(24.6) |
| New York Heart Association classification | Class | 10(5.0) | 1(0.503) | 9(4.5) |
| | Class | 23(11.6) | 2(1.0) | 21(10.6) |
| | Class | 58(29.1) | 8(4.0) | 50(25.1) |
| | Class | 108 (54.3) | 32(16.1) | 76(38.2) |
| Types of congestive heart failure | Right ventricular | 55(27.6) | 5(2.5) | 50(25.1) |
| | Bi ventricular | 64(32.2) | 12(6.1) | 52(26.1) |
| | Left ventricular | 80(40.2) | 26(13.1) | 54(27.1) |
| Etiology of congestive heart failure | Valvular Heart Disease | 51(25.6) | 16(8.0) | 35(17.6) |
| | Hypertensive heart disease | 43(21.6) | 7(3.5) | 36(18.1) |
| | Ischemic heart disease | 55(27.6) | 12(6.0) | 43(21.6) |
| | Other | 50(25.1) | 8(4.0) | 42(21.1) |

rate measurement appearing bigger at the beginning and smaller at the latter. It was seen that there is variability in pulse rate measurement between patients, indicating that random effects for each subject should be included to capture this variability and allow pulse rate measurement for patients within the same patient to be correlated (see Fig 1). From mean profile plot of pulse rate, the patients' average pulse rate was strictly decreasing until the tenth month of follow-up, after which it began to gradually decrease until the 28th month, when it changed to oscillations and began to decrease. The average pulse started to decline after the follow-up, indicating that patients with congestive heart failure were at risk at baseline. In addition, mean profile plot of pulse rate also shows that the horizontal loss smoothing approach indicates that the mean structure of pulse rate is roughly linear over time (see S1 Fig).

Fig 2 shows the variance of the pulse rate measurements of congestive heart failure patients showed an irregular pattern over the follow-up period. It increases at some point and decreases at another point, suggesting a non-constant variation among congestive heart failure patients over the follow-up period. It was observed that high variation was observed among males until

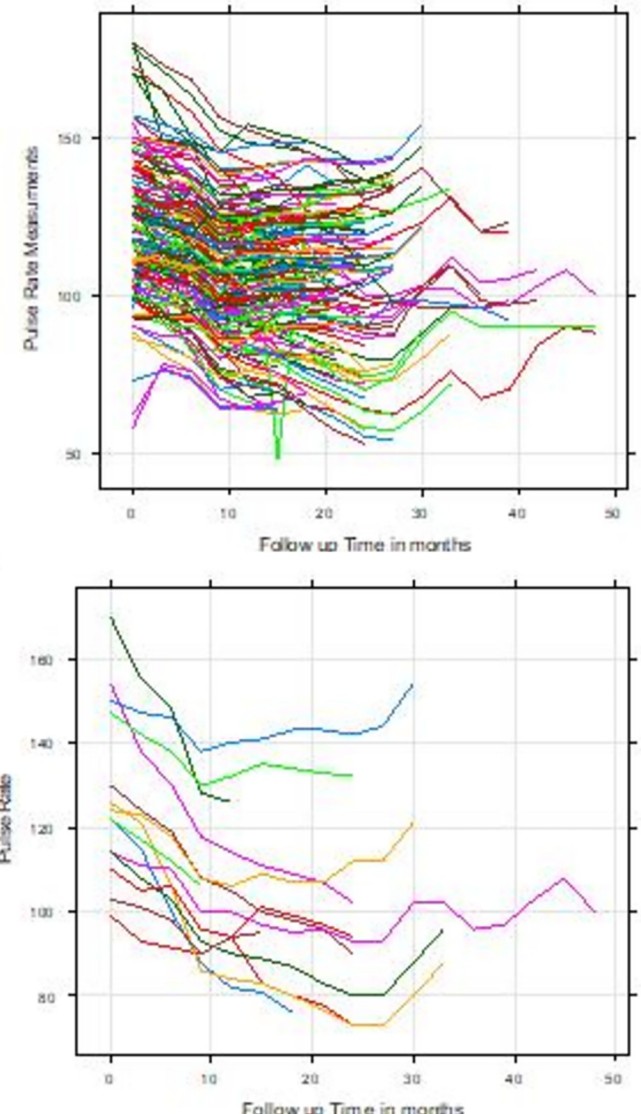

**Fig 1.**

the 18th month and after the 27th month, and the variance of both genders increased at some point and decreased at another point, which suggests there, is no constant variation over time which suggest that the mixed effect model with random intercept could be the candidate starting model to fit the data (see Fig 2).

### 3.3. Bayesian joint model of pulse with survival time

First, longitudinal measurement on the pulse rate and survival outcome time to death was separately modeled using linear mixed model and cox proportional hazard model, respectively. Accordingly, linear mixed-effect model with random intercept, random slope, and random intercept and slope were fitted and compared. Hence, the linear mixed-effect model with random intercept and slope had lower values of AIC and BIC and chosen as the parsimonious model to fit the data on the longitudinal change of pulse rate (result not shown here).

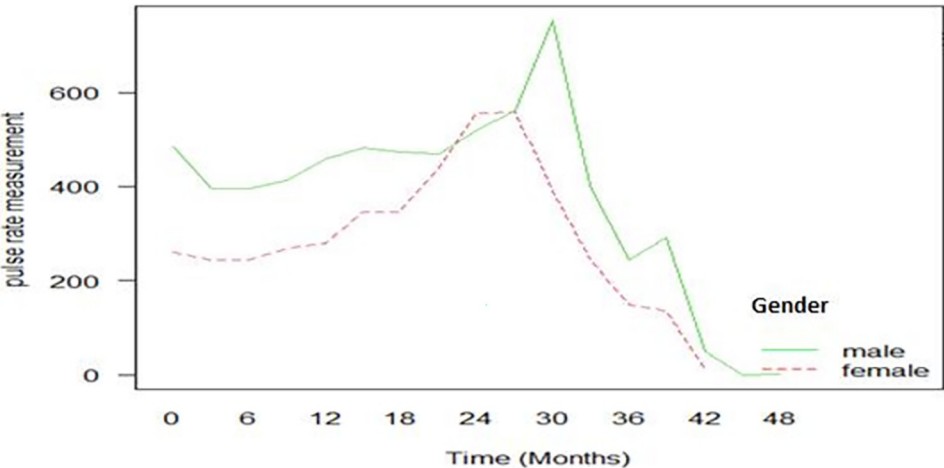

**Fig 2.**

In addition, for Cox proportional hazard model, proportional hazard assumption was tested for each categorical covariate. Hence, P-value of individual covariate and GLOBAL test statistic are greater than 5% level of significance, indicating that all covariates satisfied the proportionality assumption of the Cox model (result not shown here). Following the development of separate models, Bayesian joint model that links longitudinally measured pulse rate to the survival time to death was performed using JMbayes2 package in *R software version 4.2*. The Gibbs sampler algorithm was implemented with 20,000 iterations in three different chains initialized with over dispersed values for all parameters. Then, samples generated from the full posterior distribution are used to make inference about the joint model parameters. Before undertaking any inference from posterior distribution the convergence of generated Markov chains has been verified by density plot (Fig 3) and time series or history plot (Fig 4). Fig 3 shows density plots for only some selected statistically significant regression coefficients in the joint model and simulated samples from posterior distribution for each regression coefficient

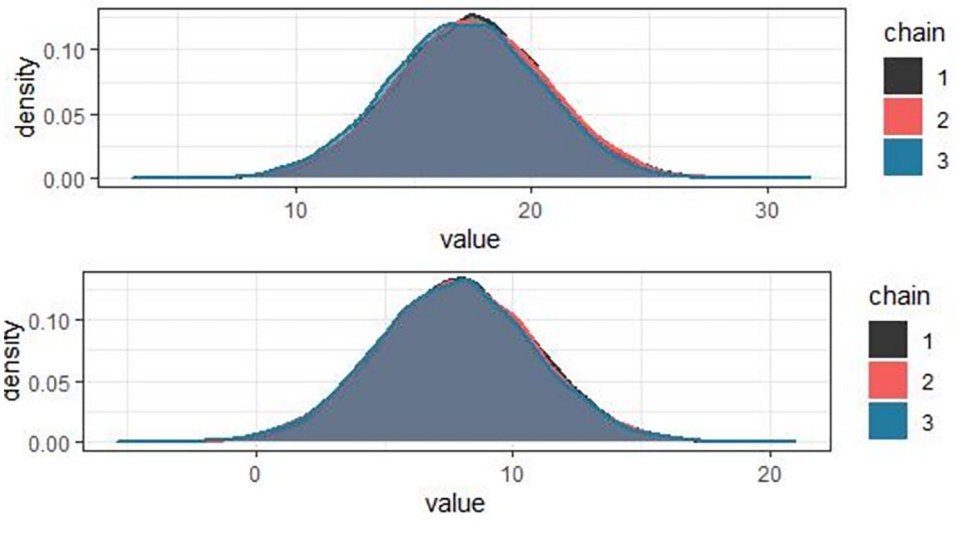

**Fig 3.**

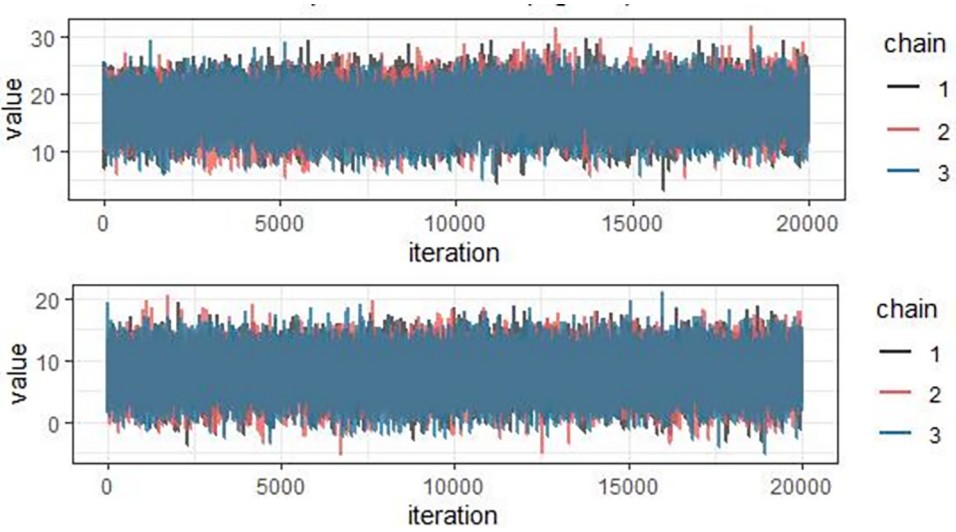

**Fig 4.**

is smooth, uni-modal shape of posterior marginal distribution indicating that simulated parameter value indicates convergence to the target distribution. The density plots for the rest of the parameters (not shown here) also tell a similar story.

Fig 4 shows history plots for only some selected statistically significant regression coefficients in the joint model and this option produces iteration number on x axis and parameter value on y-axis). The plots looks like a horizontal band, with no long upward or downward trends and the two independently generated chains demonstrated good "chain mixture" indicating that the chains has converged. The Time series (history) plots for the rest of the parameters (not shown here) also tell a similar story. This implies convergence for the regression parameters in the joint model are attained. Table 2 presents the estimates (estimated parameters, hazards ratio (HR) and 95% credible intervals for estimated posterior mean) from Bayesian joint modeling of longitudinal change of pulse rate with survival time of heart failure patients. Therefore, result from longitudinal sub model revealed that status of tuberculosis, New York Heart Association class type, left ventricular ejection fraction, gender, length of follow-up time, having chronic kidney disease, status Pneumonia, family history, status of diabetes and weight of patients at baseline had statistically significant effects on the average longitudinal change of pulse rate at 5% level of significance (95% credible interval doesn't included zero, see Table 2). Likewise result from survival sub model revealed that the covariates status of chronic kidney disease, family history, status of diabetes, smoking status of patients, status of tuberculosis, left ventricular ejection fraction, type of congestive heart failure, etiology of congestive heart failure and status of alcohol intake were associated with the time to death of congestive heart failure patients.

When the other predictors are kept constant, the average longitudinal change on pulse rate was 30.7 beats per minute for a unit increase of length of follow up time of congestive heart failure patients. Similarly, if the left ventricular ejection fraction of congestive heart failure patients increased by 1%, the average longitudinal change in pulse decreased by 0.86 beats per minute. Regarding the weight of patients at baseline, as the weight of patient increased by one kilogram the pulse rate of the patient increases by 0.15 beats per minute.

Regarding the New York Heart Association classification, the average longitudinal change of pulse rate of patient increased by 12.9 beats per minute for patient with class type IV

**Table 2. Parameter estimation of Bayesian joint model of longitudinal measurement on pulse rate and survival time to death in CHF patients.**

| Longitudinal sub-model: Fixed effects | | | |
|---|---|---|---|
| **Predictors** | **Categories** | **Posterior mean estimate of β's** | **95% CI of Posterior mean estimate of β's** |
| | Intercept | 322.2 | (167.13, 500.03)* |
| | length of follow-up time | -30.7 | (-40. 58–20.59)* |
| | Left ventricular ejection fraction | -0.86 | (-1.29, -0.47)* |
| New York Heart Association classification | Class I (Ref) | | |
| | Class II | 0.38 | (-5.76, 6.52) |
| | Class III | 5.8 | (-0.41, 12.14) |
| | Class IV | 12.9 | (3.73, 21.79)* |
| Gender | Female (Ref) | | |
| | Male | 7.6 | (1.58, 13.75)* |
| Tuberculosis status | Negative (Ref) | | |
| | Positive | 9.5 | (3.45, 15.59)* |
| Chronic kidney status | Absent(Ref) | | |
| | Present | 7.7 | (1.78, 13.78)* |
| Pneumonia status | Absent(Ref) | | |
| | Present | 17.3 | (10.99, 23.65)* |
| Family history | Absent(Ref) | | |
| | Present | 12.3 | (6.18, 18.37)* |
| Diabetes Status | Absent(Ref) | | |
| | Present | 9.4 | (3.42, 15.49)* |
| | Baseline Weight | 0.15 | (0.07, 0.24)* |
| | Baseline Age | -0.01 | (-0.06, 0.03) |
| **Random effects** | Random | Standard deviation | 95% credible interval |
| | Intercept | 374.6 | (369.8, 379.4) |
| | Slope of visiting time | 26.9 | (18.1,35.7) |
| | Corr (Intercept, slope of visiting time) | 0.14 | (0.11, 0.17) |
| Survival sub model | | | |
| Predictors | Categories | Posterior mean estimate of γ's(HR) | 95% CI of posterior mean estimate of γ's |
| | Left ventricular ejection fraction | -0.08(0.91) | (-0.12, -0.05)* |
| Etiology of congestive heart failure | Valvular Heart Disease (Ref) | | |
| | Ischemic Heart disease | -2.38(0.09) | (-4.19, -0.39)* |
| | Hypertensive heart disease | 0.99(2.69) | (-0.53, 2.56) |
| | Other | 2.84(17.16) | (1.11, 4.57)* |
| Type of congestive heart failure | left ventricular(Ref) | | |
| | Right ventricular | -3.12(0.04) | (-4.75, -1.16)* |
| | Biventricular | 0.80(2.22) | (-0.16, 1.76) |
| Tuberculosis status | Negative(Ref) | | |
| | Positive | 4.11(61.49) | (2.95, 5.26)* |
| Status of Chronic Kidney | Absent(Ref) | | |
| | Present | 2.17(8.80) | (0.93, 3.36)* |
| Smoking status | Nonsmoker(Ref) | | |
| | Smoker | 2.68(14.65) | (1.63, 3.67)* |
| Family history | Absent(Ref) | | |
| | Present | 2.77(16.02) | (1.59, 3.83)* |
| Status of Alcohol intake | No(Ref) | | |
| | Yes | 1.40(4.06) | (0.24, 2.59)* |

(*Continued*)

**Table 2.** (Continued)

| Longitudinal sub-model: Fixed effects | | | |
|---|---|---|---|
| Predictors | Categories | Posterior mean estimate of β's | 95% CI of Posterior mean estimate of β's |
| Diabetes Status | Absent(Ref) | | |
| | Present | 2.07(7.94) | (1.11, 2.97)* |
| | Association parameters(α) | 1.7 (5.47) | (1.39, 2.54)* |

Note: CI, represents credible interval, HR represents the hazard ratio and Ref denote the reference category.

congestive heart failure compared to patients with New York Heart Association class I congestive heart failure. Congestive heart failure patients with positive tuberculosis status significantly increased the mean longitudinal change of pulse rate by 9.5 beats per minute compared to patient with negative tuberculosis status.

Furthermore, regarding the gender of patients, male congestive heart failure patient had 7.6 beats per minute higher mean longitudinal change in pulse rate than female congestive heart failure patient. The congestive heart failure patients who had a family history, pneumonia, chronic kidney disease, and diabetes mellitus disease had higher average longitudinal change of pulse rate compared to congestive heart failure patients who didn't have a family history, pneumonia, chronic kidney disease, and diabetes mellitus disease, respectively. The variation of the random intercepts in pulse rate for the random portion of the linear mixed-effect model was 374.6, with a random slope of 2.69. This indicates that there is a greater baseline difference in pulse rate at the start of their treatment (see Table 2).

When the other covariates are held constant, for a unit increase in percentage of left ventricular ejection fraction risk of death in congestive heart failure patients decreased by 0.08 (HR = 0.91). That means a reduction in left ventricular ejection fraction increases the risk of mortality in individuals with congestive heart failure or vice versa. At baseline, the ischemic heart disease patient group had a hazard rate of 0.09 times higher than the Valvular heart disease patient group. Regarding the type of congestive heart failure, patients with congestive heart failure who tested positive for tuberculosis had a significantly greater risk of death (HR = 61.49). This means that patients who tested positive for tuberculosis had 61.49 times higher risk of death than who tested negative for tuberculosis.

Patients with chronic kidney disease (HR = 8.802), being smoker (HR = 14.658), having a family history (HR = 16.022), drinking alcohol (HR = 4.063), and having diabetes (HR = 7.94) were all statistically associated with higher risk of death in congestive heart failure patients compared to patients who had no chronic kidney disease, patients who were non-smoker, had no family history, non-drinking alcohol, and patients who had no diabetes respectively.

In the joint model, it was found that association parameter that link longitudinal biomarker pulse rate and the survival time to death of congestive heart failure patients was statistically significant(95% credible interval doesn't included zero, see Table 2). Hence, the estimated value of the association parameter alpha was 1.7 (HR = 5.47), indicating that the average longitudinal change in pulse rate had a positive relationship with the time to death of CHF patients across time.

## 3.4. Discussion

Properties and features of residuals, when longitudinal and survival outcomes are separately modeled have been used to check model assumptions. Result revealed that normality assumption for longitudinal process is satisfied (see S2 Fig) and in order to validate the Cox proportional hazards model assumption of the survival sub-model, a graph of the Schoenfeld

residuals was displayed to check the overall goodness of fit for survival sub-models. S3 Fig shows that the scaled Schoenfeld residuals are randomly distributed and a loess smoothed curve do not exhibit much departure from the horizontal line suggests that the proportional hazards assumption is not violated. This was also confirmed via testing the interaction of the covariate with the log of survival time (Global P-value = **0.293**). The estimated value for the association parameter ($\alpha$) in the joint model was statistically significant and indicates a strong association between longitudinal measurement of pulse rate and the risk of death. The average longitudinal change in pulse rate demonstrated a negative relationship with the length of follow-up for congestive heart failure patients. This finding is in line with conclusion from the previous study by [5]. It was found that the left ventricle ejection fraction of congestive heart failure patients had a negative significant effect on the average longitudinal change of pulse and also associated with a risk of death in congestive heart failure patients. This finding is consistent with conclusion by [25,26] and contradicts with finding by [5] which reported that left ventricular ejection fraction disease had no statistically significant effect on the risk of death. This study had no evidence of the significance of age on average change in pulse rate which contradicted the previous studies done by [5]. The baseline weight of a patient had statistically significant increasing effect on the average longitudinal change in pulse rate. This finding is consistent with previous study done by [5].

Congestive heart failure patients who have diabetes disease have a positive and significant effect on the average evolution of pulse rate and this is also related to the risk of death of congestive heart failure patients. This finding is similar to previous findings by [26,27]. Patients who had positive test result for tuberculosis had a positive significant effect on the average evolution of pulse rate and was associated with a risk of death. This result was in line with study by [26]. Furthermore, being male had a positive significant effect on the average longitudinal change in pulse rate. This finding is similar with conclusion by [26–28]. However, it contradicts previous study by [27].

## 4. Conclusions

For the joint modeling of the data, the joint model with random intercepts and slope from the longitudinal sub model and Cox proportional hazard model from survival sub model was found to be appropriate model to fit the data. Based on the result from Bayesian joint modeling of longitudinal change of pulse rate with survival time of heart failure patients, length of follow up time, *w*eight of patients at baseline, gender, chronic kidney disease, left ventricular ejection fraction, New York Heart Association class type, diabetes, tuberculosis, pneumonia and family history were statistically significant factors associated with mean evolution of pulse rate of congestive heart failure patients. From survival sub model, Left ventricular ejection fraction, Etiology of congestive heart failure type, type of congestive heart failure, chronic kidney disease, smoking, family history, alcohol and diabetes were found to be statistically significant factors associated with the risk of death of the congestive heart failure patients. In addition, computed association parameters revealed that mean evolution of pulse rate was found to be statistically significant and positively related with the hazard rate of time to death of congestive heart failure patients in the study area. To reduce the risk level, health professionals should give attention to congestive heart failure patients with high pulse rate, co-morbidities of chronic kidney disease, tuberculosis, diabetic, smoking status, family history, and pneumonia in the study area.

## Supporting information

**S1 Fig. Evolution of pulse rate for CHF patients over time.**
(TIF)

**S2 Fig. Normal probability plot of standardized residuals from longitudinal sub model.**
(TIF)

**S3 Fig. Schoenfeld residual plots of covariates: Tuberculosis and chronic kidney.**
(TIF)

**S1 Data. Supporting conclusions of the manuscript.**
(CSV)

## Acknowledgments

We thank Arba Minch University statistics department for providing permission to conduct the study via official letter of cooperation referred as stat/534/2013 with subject, To Whom It May Concern. We also extended our gratitude to staff in Arba Minch General Hospital for their kind cooperation in providing all the data for our study.

## Author Contributions

**Conceptualization:** Sebisibe Kusse Kumaso.

**Data curation:** Sebisibe Kusse Kumaso.

**Formal analysis:** Sebisibe Kusse Kumaso.

**Methodology:** Sebisibe Kusse Kumaso.

**Supervision:** Belay Belete Anjullo, Markos Abiso Erango.

**Validation:** Belay Belete Anjullo, Markos Abiso Erango.

**Writing – original draft:** Sebisibe Kusse Kumaso.

**Writing – review & editing:** Belay Belete Anjullo, Markos Abiso Erango.

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
