## [Decision Letter · Decision Letter 0]

21 Dec 2022

PONE-D-22-24580JOINT MODELING OF LONGITUDINAL CHANGE IN PULSE RATE AND SURVIVAL TIME OF HEART FAILURE PATIENTS TREATED AT ARBAMINCH GENERAL HOSPITAL, SOUTHERN ETHIOPIAPLOS ONE

Dear Dr. Anjullo,

Thank you for submitting your manuscript to PLOS ONE. After careful consideration, we feel that it has merit but does not fully meet PLOS ONE’s publication criteria as it currently stands. Therefore, we invite you to submit a revised version of the manuscript that addresses the points raised during the review process.

 The manuscript was reviewed by two experts of the field, and they have provided their comments below for you to consider and implement as much as possible. I agree with their comments and suggestions and feel that revising the manuscript based on their suggestions could improve your manuscript.

We look forward to receiving your revised manuscript.

Kind regards,

Jaimin R. Trivedi, MBBS, MPH

Academic Editor

PLOS ONE

Journal Requirements:

a) Did participants provide their written or verbal informed consent to participate in this study?

Reviewers' comments:

Reviewer's Responses to Questions

**Comments to the Author**

1. Is the manuscript technically sound, and do the data support the conclusions?

Reviewer #1: Yes

Reviewer #2: Yes

2. Has the statistical analysis been performed appropriately and rigorously? 

Reviewer #1: Yes

Reviewer #2: Yes

3. Have the authors made all data underlying the findings in their manuscript fully available?

Reviewer #1: No

Reviewer #2: Yes

4. Is the manuscript presented in an intelligible fashion and written in standard English?

Reviewer #1: No

Reviewer #2: Yes

5. Review Comments to the Author

Reviewer #1: This is a well written study. The results section needs to be abit more clear and emphasis to be made on percentages rather than frequencies. This can be re-written to make sure its reader friendly. Also, for the graphs, they need to be higher resolution, and for Figure 2, please specify the Gender categories. Table 2 would suggest remove the Ref group and put it in the footnote. Also please check on grammar and sentence structures, as some sentences are abit confusing.

Reviewer #2: The manuscript addresses an interesting topic. The data are original and the methods sound for the data at hand. The results are well described and support the conclusions. Some comments follow.

1. I really enjoy reading this work. The employed statistical methods are sound and advanced enough to model the main data features. I have just some minore notes:

a) I see that Gaussian random effects are often taken for granted, but please comment on how tenable this assumption is. Non-parametric or discrete or skew or heavy-tailed random effects can be considered instead. Of course, you use an R package, and this may be difficult to address, but some discussion is required at least.

b) Please, provide more details on the computational details, e.g. the burn-in, the priors, etc., and show the convergence of the Markov chains in a supplementary.

c) Random slopes are considered, this is fine with me. I am wondering if the model can be extended to allow for correlated random effects between the longitudinal and the survival processes. As it stands, a shared random effect approach is considered. Some examples are available in the statistical literature, see e.g. papers published in Statistics in Medicine or Biometrical Journal.

d) The random effects are time-constant. This is a limitation of the modelling. Nowadays, time-varying random effects are commonly used in longitudinal data analysis, and also in the joint modelling with survival processes. I am wondering why you did not allow the random effects to vary over time, e.g. following a Markov chain or an Autoregressive process. A discussion on this point is also required.

2. No matter which approach is considered, the assumptions of which the methods are based must be checked carefully. This aspect is overlooked. A residual analysis should be provided.

3. Collinearity may be an issue, as well as endogeneity. How do you deal with these potential problems affecting the results?

6. PLOS authors have the option to publish the peer review history of their article (what does this mean?). If published, this will include your full peer review and any attached files.

Reviewer #1: No

Reviewer #2: No

---

## [Author Response · Author response to Decision Letter 0]

14 Jan 2023

Dear Reviewers,

We thank the reviewers for the kind words on our manuscript and the points brought forward as they resulted in an improvement of the submitted manuscript. The manuscript has been improved according to the suggestions of the both reviewers as detail given in rebuttal letter.

---

## [Decision Letter · Decision Letter 1]

20 Feb 2023

JOINT MODELING OF LONGITUDINAL CHANGE IN PULSE RATE AND SURVIVAL TIME OF HEART FAILURE PATIENTS TREATED AT ARBAMINCH GENERAL HOSPITAL, SOUTHERN ETHIOPIA

PONE-D-22-24580R1

Dear Dr. Anjullo,

We’re pleased to inform you that your manuscript has been judged scientifically suitable for publication and will be formally accepted for publication once it meets all outstanding technical requirements.

Kind regards,

Jaimin R. Trivedi, MBBS, MPH

Academic Editor

PLOS ONE

Additional Editor Comments (optional):

Reviewers' comments:

Reviewer's Responses to Questions

**Comments to the Author**

1. If the authors have adequately addressed your comments raised in a previous round of review and you feel that this manuscript is now acceptable for publication, you may indicate that here to bypass the “Comments to the Author” section, enter your conflict of interest statement in the “Confidential to Editor” section, and submit your "Accept" recommendation.

Reviewer #1: All comments have been addressed

Reviewer #2: All comments have been addressed

2. Is the manuscript technically sound, and do the data support the conclusions?

Reviewer #1: Yes

Reviewer #2: (No Response)

3. Has the statistical analysis been performed appropriately and rigorously? 

Reviewer #1: Yes

Reviewer #2: (No Response)

4. Have the authors made all data underlying the findings in their manuscript fully available?

Reviewer #1: Yes

Reviewer #2: (No Response)

5. Is the manuscript presented in an intelligible fashion and written in standard English?

Reviewer #1: Yes

Reviewer #2: (No Response)

6. Review Comments to the Author

Reviewer #1: (No Response)

Reviewer #2: (No Response)

7. PLOS authors have the option to publish the peer review history of their article (what does this mean?). If published, this will include your full peer review and any attached files.

Reviewer #1: No

Reviewer #2: No

---

## [Editor Report · Acceptance letter]

24 Feb 2023

PONE-D-22-24580R1 

*Joint modeling of longitudinal change in pulse rate and survival time of heart failure patients treated at Arbaminch General Hospital, Southern Ethiopia*

Dear Dr. Anjullo:

I'm pleased to inform you that your manuscript has been deemed suitable for publication in PLOS ONE. Congratulations! Your manuscript is now with our production department. 

Kind regards, 

on behalf of

Dr. Jaimin R. Trivedi 

Academic Editor

PLOS ONE